# Policy Changes in China’s Family Planning: Perspectives of Advocacy Coalitions

**DOI:** 10.3390/ijerph20065204

**Published:** 2023-03-15

**Authors:** Zhichao Li, Xihan Tan, Bojia Liu

**Affiliations:** 1School of International and Public Affairs, Shanghai Jiao Tong University, Shanghai 200030, China; 2School of Public Administration and Policy, Renmin University, Beijing 100872, China; 3School of Political Science and Public Administration, East China University of Political Science and Law, Shanghai 201620, China

**Keywords:** policy change, family planning policy, advocacy coalition framework, preferential attachment, discourse network analysis

## Abstract

Studies on policy change focus on governmental decision-making from a technical rationality perspective, ignoring the fact that policy change is a complicated social construction process involving multiple actors. This study used the modified advocacy coalition framework to explain changes in China’s family planning policy and discourse network analysis to show the debate on the birth control policy among multiple actors (central government, local governments, experts, media, and the public). It found that the dominant coalition and the minority coalition can learn and adjust deep core beliefs from each other; the sharing and flow of actors’ policy beliefs drive change in the network structure; and actors’ obvious preferential attachment when the promulgation of the central document, are all helpful in policy change. This study can explain macro-policy changes from a micro-perspective to reveal the process and mechanism of policy changes in China’s authoritarian regime.

## 1. Introduction

Policy change is a classic topic in the fields of political science and public management, and many studies on policy change have investigated the types, determinants, characteristics, and impacts of change [1,2,3]. China’s policy change in family planning has attracted increasing attention because of China’s rise as a superpower and its growing influence [4]. Many studies have used Western theoretical models to test China’s policy directly, which is beneficial at the macro-structural analysis level of policy change [5,6]. However, in the authoritarian regime, the inequality of power relations among the multiple actors involved in policy changes is self-evident [7]. The question is, how does large-scale policy change occur in China?

China’s family planning policy was determined as the basic national policy in September 1982, when it was written into the Constitution. The policy was strictly implemented in China for over 30 years. However, recently, policies such as the universal two-child policy have been introduced, indicating that the one-child policy has been formally phased out. Why has the family planning policy changed rapidly? What are the mechanisms that give rise to such policy change, and how has the process evolved? If completely attributed to the central authorities, then the policy change can be regarded as a ‘black box’ or simply summarised as a rational decision; that is, the central government has adjusted the policy based on the dramatic change in population structure. This explanation not only overstates the power of the central government, even in an authoritarian regime such as China, but also oversimplifies the policy change process. Therefore, this study attempts to use the modified advocacy coalition framework to present this complicated process more concretely and accurately.

The advocacy coalition framework (hereinafter the ACF) was established to examine policy changes more than a decade ago [8]. In terms of a generalisable theory of policy change, the ACF shows advantages in its portability in different environments, particularly across North America and Western Europe [9,10]. However, public policy research has faced numerous challenges over time. These challenges partly stem from the dilemma of developing common concepts and conducting research across time and space [11,12,13]. The ACF research has few applications outside of Europe and America [14]. Therefore, the development of the ACF across time and space is an important goal of public policy research, which requires an understanding of why certain political choices or coalitions arise in particular situations, and the ACF is a general and flexible way of explaining different national or regional contexts.

Although the ACF seemingly provides one of the most promising and influential approaches for understanding policy change [15,16], it is still difficult to reflect micro-changes; that is, how the dynamic changes in members’ policy beliefs affect advocacy coalitions and how these coalitions then affect policy change. The ‘linguistic turn’ of Western public policy research provides a method that overcomes the shortcomings of the ACF [17] by regarding discourse as the product of value intention and can change social reality. When studying various languages or text materials, discourse researchers no longer regard them as a purely ‘descriptive’ or ‘reflective’ sign system of the objective world but uphold the position of constructivism philosophy, believing that they can fundamentally ‘constitute’ or ‘change’ social reality. Therefore, scholars have proposed discourse network analysis (DNA), arguing that the process of policy change occurs when some advocacy coalitions share similar ideas and compete for discourse influence [18,19]. DNA provides a basic perspective for both the analysis of ‘discourse’ and its impact on policy change.

Therefore, to show the micro-process of China’s policy change, this study takes the family planning policy in China as an example, introduces the modified ACF, and uses DNA to explore the formation and evolution process of different advocacy coalitions based on the diverse policy beliefs of multiple actors (central government, local government, experts, the media, and the public). It also provides a mesoscopic perspective to understand and analyse China’s large-scale policy change.

This paper is organised as follows. Section 2 reviews the ACF literature and policy change theory, and it introduces the family planning policy. Section 3 introduces the data source, coding process, and two analysis methods. Section 4 presents the empirical results, and Section 5 presents the discussion and conclusions.

## 2. Understanding the Modified ACF in China’s Family Planning Policy

### 2.1. The Modified ACF and Policy Change

The ACF was proposed by Sabatier and Jenkins-Smith in the mid-1980s to provide a conceptual framework for explaining policy change, which may include changes in external and internal subsystem events [16], changes through policy-oriented learning [20], and changes in coalitions that reach an agreement through negotiation when faced with a policy stalemate [21]. Although various studies have confirmed that ACF can be used to verify the policy change of authoritarian regimes, the policymaking process in China is no longer controlled solely by state actors but does involve non-state actors [6,22,23]. Therefore, we still need a modified ACF that is more suitable for China’s national conditions. As Figure 1 shows, the modified ACF captures the ongoing interactions around the policy subsystem, the political use of preferential attachment, conflicting or coordinated beliefs, and strategies of actors inside and outside the central government in response to policy changes.

Advocacy coalitions exist in the policy subsystem, and usually, there are two advocacy coalitions composed of policy actors from different levels who compete with each other and have a common belief system within a particular policy subsystem [24]. However, unlike Western governments, opposition coalitions probably did not exist in China because of the Party’s dominance. As a unitary political context, policy change exists in multiple governmental ministries and agencies relevant to China’s policy process [25]. In China, government departments, including the central and local governments, hold the authority of public policy and occupy a core position [26]. Although non-authoritative power advocacy coalitions, such as minority coalitions formed by experts, media, and the public, have played a certain role in promoting public policy changes [23,25], they usually live on the periphery of the policy subsystem and generally cannot directly lead to public policy changes.

**Hypothesis** **1:** **(Policy Change)**
*There are at least two advocacy coalitions in China’s policy change, a dominant coalition led by the government and a minority coalition composed of multiple actors such as experts, media, the public, and so on.*


Faced with alternative insights and criticisms put forward by competitors sharing new experiences, new information, and technological knowledge, coalition members will absorb reasonable factors from competitors and make certain adjustments to their belief systems, known as policy-oriented learning. This belief system can be understood as a conceptual system that has three levels: deep core beliefs, policy core beliefs, and secondary beliefs [24]. However, once China’s major policies enter the implementation stage, due to the complexity of the national and social conditions, the interests of various localities, departments, organisations, groups, and even individuals will gradually solidify around the current policies, making it difficult for the specific interest groups to adjust deep core beliefs, let alone to adjust stable policies such as family planning policy. Experts, media, and the public generally aggregate in minority coalitions to form social pressure on the dominant coalitions by delivering expert opinions, joining academic forums and using media exposure, and so forth. In China, the central government is the main decision-maker for policy formulation and policy changes. Once the practice of birth control does not adapt to environmental change or the seriousness of population problems causes social–economic problems that threaten the Party’s survival, the central government will promptly adjust the policy, break the original stable policy, and change its deep core beliefs [27].

**Hypothesis** **2:** **(Policy Change)**
*China’s advocacy coalitions are not equal in power, but the dominant coalition and the minority coalition can learn and adjust deep core beliefs from each other, which promotes policy change.*


Preferential attachment has been recognised as a potential self-organisation mechanism of network formation, and it is related to the theory of the Matthew effect that the nature of preferential attachment is the principle of ‘the rich get richer’ or, more generally, ‘cumulative advantage’ [28,29]. Thus, based on this principle, we could define the preferential attachment from the perspective of existing nodes. It states that the more existing ties one node has, the newer connections it is likely to accumulate [30]. In this study, preferential attachment infers that the ability to gain policy signals released by the central government may increase with the actor centralities in the network. Due to China’s top-down administrative hierarchy, multiple actors will move close to the central government naturally after a document is released. However, although the central government has strong political power in the implementation of existing policies, the central government’s policy discourse is not the final step, and local governments still have considerable autonomy to show the co-existence of preferential attachment and independent ideas in policy discourse in the process of policy implementation [31,32].

**Hypothesis** **3:** **(Policy Change)**
*Multiple actors show a different degree of preferential attachment once the central document promulgates.*


### 2.2. Application of the Modified ACF to China’s Family Planning Policy

The ACF must meet the following four assumptions to view policy change: (1) the observation time of policy change should be at least 10 years; (2) the most effective way to consider policy change is to focus on the policy subsystem; (3) the policy subsystem needs to cover all levels of government, consultants, scientists, and the media; and (4) public policy can be conceptualised in the same way as a belief system [8]. China’s family planning policy is a basic national policy that has been implemented for more than 30 years at all levels of the Chinese government and meets all the above assumptions. The family planning policy has been carried out for more than three decades; however, in 2013, the one-child policy was modified to include couples where both a husband and wife from a single-child family can have two children (Dandu Er’tai policy). In 2015, the Chinese Communist Party (CCP) decided to adopt the two-child policy. Why did such a strong family planning policy change on a large scale? What was the logic behind this change?

The modified ACF tries to answer these questions. The central feature of the modified ACF is the concept of advocacy coalitions. To correctly apply the ACF in China’s political system, it is necessary to investigate the family planning policy, which has at least two advocacy coalitions in the policy subsystem [33]: the dominant coalition, whose deep core belief is adhering to the family planning policy, and the minority coalition, whose deep core belief is advocating adjusting the policy. The dynamic game between the two coalitions promoted the process of policy change of the family planning policy.

Considering the modified ACF still cannot intuitively reflect how the coalition members and their beliefs change, this study used an analytical tool–discourse network to make up for this shortcoming. Western public policy research has emphasised the discourse factor of policy since 2000, influenced by ‘discursive turn’ [17] and ‘narrative turn’ [34]. Brandes et al. first attempted to incorporate discourse into social network analysis and demonstrated the process of dynamic discourse visualisation through dynamic centring resonance analysis [35]. Later, Leifeld introduced the discourse factor into policy network theory and proposed DNA, which can, to a certain extent, remedy the shortcomings of the modified ACF by explaining the interaction among actors from a quantitative perspective [36]. DNA reveals the relationships between actors in terms of policy beliefs, as well as the relationships between actors belonging to a specific coalition who share the same policy beliefs. Simultaneously, based on the principles of social network analysis, actors’ interactions within the policy subsystem can be visualised and deconstructed dynamically.

Therefore, applying the modified ACF to China provides an opportunity to understand the extent that ACF’s theories about advocacy coalitions, policy-oriented learning, and policy change, as developed in the context of Western democracy, can be applied to an authoritarian political system. China’s family planning policy was a process of centralisation and a typical example of an autocratic regime [37]. To explore why there was rapid and significant policy change after long-term policy stability, this study tested the applicability of the modified ACF in the context of authoritarianism. To some extent, policy change in China has become more open after the transition from a planned economy to a market economy, which is the combination of an authoritarian political system and economic structural reform that makes it a valuable test to synthesise and understand the various applications of the modified ACF in China [22,25]. This study attempted to combine the modified ACF with DNA to analyse changes in family planning policy, where discourse concepts can be regarded as policy beliefs. This not only clarifies the discourse evolution of multiple actors behind the policy but also provides a more reliable quantitative empirical analysis method for the study of China’s large-scale policy change.

## 3. Method

### 3.1. Data Source and Coding

The data were mainly from the Chinese Core Newspaper Database, which contains 632 important newspapers published in China since 2000. Extant research has indicated that newspapers are sources of multiple actors’ voices and that they reflect more than the views of the publishers since they must learn to negotiate between the lines of political correctness, economic profitability, and social responsibility [25,38]. The newspaper selection considered two aspects. First, the examination of whether the adjustment of the family planning policy was ultimately decided by the central government was considered. Therefore, the following four representative Central Party newspapers were selected: The People’s Daily, Guangming Daily, Economic Daily, and Xinhua Daily Telecommunications. Second, the family planning policy has obvious demographic characteristics; therefore, three social science representative newspapers were chosen: China Population News, Chinese Social Science News, and Social Science News.

This study drew on Strauss’ three-stage analysis of grounded theory to acquire the discourse concept of actors through open, axial, and selective coding [39,40], as follows:(1)Open coding is the process of conceptualising and categorising actors’ policy beliefs. First, this study’s period of analysis was set from January 2004 to December 2015 in the database, and the keyword ‘the family planning policy’ was searched. In total, 1157 texts were obtained. Second, coders read each text carefully and identified the actors therein, that is, the multiple actors who had discourse expressions on family planning. Then, the actors’ policy beliefs were encoded.(2)Axial coding clusters of the discourse concepts were selected by open coding to form 22 secondary beliefs that supported the dominant coalition and 28 secondary beliefs that supported the minority coalition. Thus, the two-mode network ‘actor concept’ was created. When an actor mentioned a discourse concept, the concept was given a value of 1; otherwise, the value was 0.(3)Fifty axial codes were linked to verify the relationship between concepts, and the concepts that had not been specified were supplemented. Finally, three selective codes—the policy core beliefs—were established: the seriousness of population problems, causes of population problems, and advocating plans.

Coders used NVivo 11 to encode 1157 texts, identify 387 actors, and cluster 50 axial codes. After calculating reliability, Cohen’s kappa index of the codes was 0.88. Therefore, the coding work had high reliability and met the needs of DNA.

### 3.2. DNA

DNA is a discourse measurement and visualisation tool based on discourse text classification and social network analysis. In addition, it is also an analytical tool under the ACF that can identify sub-advocacy coalitions, thus opening the complex discourse structure quantitatively [41]. DNA usually contains actors and concepts, and this study established a link between the two from the following steps.

The first step constructed an affiliation network, namely, a two-mode network based on common discourse concepts. Figure 2 is an illustrative example showing that when actor a1 proposes concept c1, a1 is connected to c1. When actor a2 proposes c1 again, a2 is connected to c1.

The second step built an actor or concept network. The actor network builds a one-mode network with actors as nodes and concepts as edges. The concept network builds a one-mode network with concepts as nodes and actors as edges. The actor network can be divided into actor and actor conflict networks. For the former, when two actors hold a similar attitude towards a certain concept, they form an actor congruence network. If two actors hold the same concept more strongly, they are more likely to belong to the same advocacy coalition. For the latter, when two actors hold opposing opinions on the same discourse concept in the network, they form an actor conflict network. This study chose the actor congruence network. When two actors hold the same attitude towards the same concept, they establish a connection (Figure 3).

The third step visualised the network using Python. Actor nodes were divided into five categories in the network (see Figure 4, Figure 5 and Figure 6): national leaders, central government agency personnel (squares), local government agency personnel (circles), experts (upwards triangles), media (right triangles), and the public (inverted triangles). The Louvain algorithm automatically clustered nodes and assigned a colour to each node. The closer the colours were, the greater the probability that the nodes belonged to the same faction. The size of a node was determined by the degree of centrality.

### 3.3. Exponential Random Graph Model

The exponential random graph model (ERGM) is a statistical model that describes the connection relationship between nodes and the local structure of the network [42]. The ERGM helps to examine the attribute characteristics of a network node, including basic demographics, economic factors, social environments, and other factors, as well as the effect of the attribute characteristics on the network structure. Creating and sustaining policy networks is a strategy for political actors to exchange resources and information and try to influence policy decisions. In this regard, the network structure enables ERGM to reveal the role of the observed specific dependency structuref in the network formation process and explain the existence of ties in the networks. In the practice of family planning discourse, central discourse inevitably occupies an important position. Nevertheless, the local government’s preference for central policy is not always dependent, as they have discourse autonomy. The significance of using the ERGM is being able to clarify the influence of multiple actors’ legal rights on the structure of the discourse network during different stages of policy change and present the micro-process of policy change.

Specifically, this study used the main effect model of the ERGM through R. This model compares the probability of connection between the reference node and other nodes by using an attribute as the benchmark. If other nodes are significant, the probability of connection between the attribute and the benchmark is high [43]. In the discourse network, if the probability of connection between a particular attribute and other nodes is high, this attribute node shares more discourse with other attribute nodes. Taking the central government discourse as the benchmark, this study used the main effect model to compare and analyse the behavioural changes of other discourse actors in the discourse network of family planning.

## 4. DNA from the One-Child Policy to the Universal Two-Child Policy

China’s family planning policy was implemented with high rigidity and sustainability and successfully realised the transformation of population reproduction for 30 years. In the 1990s, the population showed new characteristics of a low birth rate, death rate, and overall growth rate. However, the population characteristics gradually changed in the 21st century. The negative effects of the family planning policy, such as the ageing population, disappearance of the demographic dividend, and imbalance of the sex ratio at birth, were increasingly significant, leading to debate among government officials, scholars, the media, and the public. Under this circumstance, the central government considered putting adjustments to the family planning policy on the agenda. A critical period was between 2004 and 2015 for family planning policy changes. During this period, the 2013 Dandu Er’tai policy and the 2015 universal two-child policy were the key nodes of the updated family planning policy. What was the driving mechanism behind the change in family planning policy? What was the micro-theoretical basis of this policy change? This study divided the family planning policy into three stages: discourse diversion, bipolar discourse, and discourse confluence.

### 4.1. Discourse Diversion Stage: Formation of the Minority Coalition (2004–2009)

The minority coalition emerged after a period of research and policy-oriented learning by experts appointed to the National Family Planning Commission. With a sharp decline in the total fertility rate, demographers started to question whether containing birth control should be a primary policy goal. In 2004, Professors Gu Baochang and Wang Feng established a research group on China’s fertility policy in the 21st century, and 18 population experts jointly drafted and signed a proposal to adjust the policy [44]. This proposal concerned the central government and relevant departments and was the start of discourse change for the family planning policy. However, the decision on ‘Comprehensively Strengthening Population and Family Planning to Solve Population Problems’ made by the CCP and the State Council in 2006 showed that the family planning policy was not loosened. Although the central government’s attitude towards the family planning policy did not change, the call for adjusting it gradually appeared in academic circles, which promoted the formation of the minority coalition.

Figure 4 shows the actor congruence network from the period 2004–2009, in which the outer lines show the connections between different nodes. As we can see in Figure 4, the red part represents the dominant coalition, while the blue represents the minority coalition, and the yellow shows the other coalition. To explore the local characteristics of the network, this study explored faction analysis using the tabu search algorithm to optimise and measure the function of the faction partition, with the purpose of determining a specified number of factions.

According to Figure 4 and Table 1, there were three advocacy coalitions from 2004 to 2009. Experts from the fields of demography and sociology were the primary voices in the minority coalition. Many experts appointed to the NFPC suggested their own approaches for adjusting the family planning policy, such as Professor Tian Xueyuan’s ‘Three Population Forecasting Options’, Professor Zhai Zhenwu’s ‘Three-Step Plan’, and adjustment opinions of Professor Zeng Yi et al. [45,46,47]. The dominant coalition is represented by the central and local governments. The media were included in the ‘other coalition’ category. The diagonal data in Table 1 show the internal relationship of each advocacy coalition. The inner subgroup density of the dominant coalition is the highest (0.873), followed by that of the minority coalition (0.722), which means that actors communicate closely with their own coalition members. However, there is a lack of discourse communication between different advocacy coalitions.

To measure the degree of agglomerated subgroups more accurately, the E-I index was introduced. The value range of the E-I index is [−1, 1]. The closer the value is to 1, the more likely the relationship is to occur outside the subgroups, which means the smaller the number of factions; the more the value tends to −1, the more the relationship tends to occur within the subgroup, and the greater the number of factions; the closer the value is to 0, it indicates that the relationship tends to be randomly distributed and the factions are not obvious. The formulation of the E-I index is as follows:


$$\mathrm{E-I}\;\;\mathrm{index}=\frac{\mathrm{Density}\;\mathrm{of}\;\mathrm{subgroups}}{\mathrm{Overall}\;\mathrm{density}}$$


From the period 2004–2009, the E-I index of the entire network is 0.196, which shows that many subgroups exist. The E-I index of the dominant coalition is −0.712, and the number of internal relations is far greater than those of external relations, except for other actors, indicating that the internal relations of the dominant coalition are close. The E-I index of the minority coalition is −0.213, and the internal communication between the coalition is greater than that of external communication. The results in Table 2 are, thus, confirmed again.

Degree centrality is used to analyse the number of nodes connected to a node in the network, which can be regarded as the measurement of the policy belief’s universality of an actor in the discourse network. An actor with more relationships has a larger degree of centrality. Table 3 shows the result of the degree centrality analysis of the network from 2004 to 2009, in which degree centrality reflects the dominant position of network nodes. This study selected the top 15 actors in the overall network and found that all belonged to the dominant coalition. Specifically, local governments such as Liaoning, Jiangsu, and Shandong were in the top 5, respectively. Central government agencies such as NPFPC and government officials such as Hu Chunhua and Zhang Weiqing also had a high degree of centrality. The results suggested that the government played a leading role in the family planning policy change from 2004 to 2009.

### 4.2. Discourse Bipolar Stage: Debate between Dominant Coalition and Minority Coalition (2010–2012)

Following research from 2005 to 2008, the research group on ‘China’s fertility policy in the 21st century’ put forward Suggestions on the Adjustment of China’s Population Policy in 2009. This proposal made some government actors change their attitudes and reflect on the applicability of the one-child policy, and it stimulated an increasing number of actors to participate in the policy debate [45,48].

Figure 5 shows two obvious advocacy coalitions. Most experts occupy the right subgroups, while a large number of local governments occupy the left subgroups, and the central government is widely distributed across the two subgroups. To further explore the relationship between subgroups, faction analysis and E-I index analysis were conducted (Table 4).

There are only two advocacy coalitions on behalf of the minority and the dominant coalitions, respectively, while the ‘other coalition’ subgroup disappears. A possible explanation is that some actors from the ‘other coalition’ subgroup held a wait-and-see attitude from 2004 to 2009. The media played its role as an intermediary at this stage, transmitting views between the two major coalitions and forming their policy beliefs through policy learning over time, and joined the debate of the two major advocacy coalitions from 2010 to 2012 (Table 5).

The E-I indices of the minority and dominant coalitions are −0.529 and −0.606, respectively, and both are less than 0, indicating that the internal communication of the two advocacy coalitions is extremely strong and the external communication is weak. The E-I index of the whole network is −0.571, which shows a significant number of subgroups and that the two advocacy coalitions maintain polarisation.

From 2010 to 2012, the minority coalition was mainly composed of experts, scholars, and the public, and they expressed their appeal to the central government through the media. Simultaneously, the central government’s attitude towards maintaining the family planning policy was tempered. In 2010, the National Population and Family Planning Commission stated that the Dandu Er’tai policy pilot work should be promoted in the Twelfth Five-Year Plan for National Population Development (Consultation Draft). In 2011, this plan was submitted to the State Council, which was about to release an updated policy.

However, at this stage, the dominant coalition, composed of local governments, opposed the loosening of the family planning policy. For example, the local population and family planning committees in many provinces and cities held negative opinions on slackening the fertility policy because they were worried about the rebound of the fertility rate and the pressures on limited resources and environmental pollution. With the opposition voice being stronger, the Party leaders were unsure about the consequence of policy relaxation. Therefore, the process of the Dandu Er’tai policy was set aside on account of the important period of the convening of the 18th National Congress of the CCP in 2012 and the subsequent government institution reforms of the State Council [49]. The minority coalition and dominant coalition, thus, entered a state of confrontation.

### 4.3. Discourse Confluence Stage: Combination of Dominant Coalition and Minority Coalition (2013–2015)

In March 2013, the National Population and Family Planning Commission and the Ministry of Health merged to establish the National Health and Family Planning Commission of the People’s Republic of China in the new round of government institutional reforms of the State Council. The change in organisational structure signalled a shift in policy. In Figure 6, although there are still two major coalitions, some of the governments are already in the transitional stage, as the yellow part shows. The confrontation between the two coalitions disappears, and multiple actors are prone to merge.

Table 6 shows the E-I index analysis. The E-I index of the dominant coalition is 0.480, and the external relationship index is larger than the internal relationship index. Moreover, the E-I index of the minority coalition is −0.865. The internal communication of the coalition is close, sharing a vast majority of discourses. In Figure 6, the discourse of the National Family Planning Commission, which once belonged to the dominant coalition, changes and joins the same subgroup as one of the members of the minority coalition.

During the 2013 National People’s Congress and Chinese People’s Political Consultative Conference, some representatives, for example, Wu Shimin, proposed gradually loosening the family planning policy [50]. On 12 November 2013, the Central Committee of the CCP passed the Decision to Deepen Reforms on Several Major Issues and formally proposed the implementation of the Dandu Er’tai policy. Thus, the family planning policy changed substantially for the first time.

The introduction of the above policy spawned an urgent call for the implementation of the universal two-child policy. In 2014, a joint proposal letter from 5000 couples asking for the universal two-child policy became a landmark event in the national attitude toward this topic [51]. Subsequently, on 29 October 2015, the Central Committee of the CCP promulgated a policy that a couple could have two children. The universal two-child policy, issued by the Amendment to the Population and Family Planning Law (Draft), officially marked the end of the one-child policy and began the era of a relatively loose fertility policy in China.

### 4.4. Discourse Concept Analysis

Discourse can reveal stories [52], and the flow of discourse concepts can directly reflect the track of policy change. Therefore, by constructing a conceptual network of three stages, this paper obtains the degree of centrality of each node in each stage, with a view to showing the evolution of discourse concepts in different stages of policy change. The greater the degree of centrality of network nodes, the more times the concept is shared by actors, and the more important this concept is at this stage.

In Table 7 and Table 8, it can be seen that there were dynamic continuities among different policy beliefs at different stages in both coalitions. The motivational basis of the policy change of the two coalitions in different stages can also be found in the changing degree of centrality of the policy beliefs, especially the secondary beliefs. The overall results of degree centrality in Table 7 and Table 8 show that there existed an upward trend for the degree of centrality of the discourse concept of the minority coalition while it showed a downward trend of the dominant coalition, which partially presented the transition of family planning policy from policy stable to policy change. Through discourse analysis of 22 policy beliefs of the dominant coalition in Table 7 and Table 8 policy beliefs of the minority coalition in Table 8, the following conclusions can be drawn.

(1) At the discourse diversion stage, the minority coalition began to call for adjustment of the family planning policy because of population problems. The method for adjustment was to advocate the Shuangdu Er’tai policy. However, the dominant coalition insisted on the basic national policy to stabilise the low fertility level of the population and improve its quality.

(2) At the discourse bipolar stage, the minority coalition called for the adjustment of the family planning policy and proposed the implementation of the Dandu Er’tai policy. Some actors even suggested cancelling the family planning policies. However, the dominant coalition was limited by the dilemma of China’s large population base, as well as great pressures on resources and the environment; hence, this coalition adhered to the family planning policy.

(3) At the discourse confluence stage, the degree of centrality of the discourse concept of the minority coalition was generally higher than that of the dominant coalition. The dominant coalition’s policy beliefs, such as a ‘large population base’ and ‘fertility rate rebound’, gradually converged, and the implementation of the universal two-child policy was expected. Nonetheless, at the same time, the ‘no three-child’ and ‘basic national’ policies were still highly degree-centred, which shows that the adjustment of family planning policy in China was still limited and conditional.

## 5. Preferential Attachment in Policy Change

Due to the presupposition of equality of all policy subjects in the ACF theory, the legal rights dimension of multiple discourse actors had not been reflected in the ACF framework. However, from the analysis of China’s policy change process, legal rights are an indispensable dimension that plays an important role in political order and policy decision-making. Therefore, to present the micro-process of policy change, this study introduced the ERGM to clarify the influence of multi-actors’ legal rights attributes on the evolution of discourse network structure during different stages of the policy change.

Table 9 shows the results of the three-stage discourse network in the main effect model (taking the central government as the base), reflecting the degree of discourse connection of multiple actors during different periods. From 2004 to 2009, various actors are linked to the central government. To further explore the relevant characteristics of each actor’s policy beliefs compared with those of the central government, the following density formula is introduced:$$P=\frac{1}{1+e^{-\left(\theta_{1}X_{1}\right)}}$$
where $\theta_{1}$ represents the coefficient of the edge statistics and *X*_1_ represents the statistics of the change with the number of sides as the variable. Because the probability of forming an edge is consistent with the density of the network, the density formula means that the higher the density of the network, the greater the probability of communication between actors. The results show the network density of local governments and the central government (0.549), the media and central government (0.43), experts and the central government (0.381), and the public and central government (0.341). Therefore, from 2004 to 2009, the interaction between local governments and the central government is the strongest, and local governments have the highest degree of preferential attachment to the central government.

From 2010 to 2012, there was a period of intense debate between the minority coalition (with experts as the key actors) and the dominant coalition (with the local governments as the key actors). Both coalitions attempted to convey their policy beliefs to the central government, which is also reflected in the main effect model (2010–2012). By calculating the network density, compared with experts, local governments maintain a stronger discourse contact with the central government (P_local governments_ = 0.444 > P_professors_ = 0.418).

From 2013 to 2015, only the public has a significant connection with the discourse of the central government. This was probably because the dominant coalition had policy-oriented learning from the minority coalition. Especially after the central government solicited public opinion on the universal two-child policy, the public shared more policy beliefs with the central government. The changes in the family planning policy in 2013 encouraged bottom-up advocacy by ordinary citizens. Due to the development of social media, the public’s response and support for central policy became the strongest voice in the discourse network. Therefore, the policy change of 2013 also provided an opportunity for the central government to evaluate the effects of the relaxation of birth control. The direction of the family planning policy had been very clear because of the issuance and implementation of the policy document; therefore, apart from the public, there was no need for more actors to speak for the adjustment of family planning policy and the connection to the central government was no longer significant.

## 6. Discussion and Conclusions

This study attempted to use the modified ACF to explain China’s large-scale policy change by analysing the family planning policy, which defined two advocacy coalitions: the authoritative and minority coalitions. Moreover, it sorted 50 secondary beliefs of the policy subsystems and portrayed the dynamic evolutionary trajectory of the discourse network from multiple actors.

This study has two key findings. First, it shows how China’s large-scale policy changes occurred and confirms the Hypothesis 1 and 2. The change in the family planning policy can be divided into three stages: diversion, polarisation, and confluence. The actors outside the government put forward policy changes according to their policy beliefs and were included in the government agenda. In the first stage (2004–2009), when the dominant coalition and minority coalition were initially formed, the dominant coalition, backed by the authorities, played a leading role; therefore, the family planning policy remained unshakeable. In the second stage (2010–2012), there was a substantial divergence between the minority coalition, mainly supported by experts, and the dominant coalition, mainly supported by local governments. Although there was no transformational change, some indications showed that the family planning policy was going to change. For the third stage (2013–2015), due to the rising voice posed by the minority coalition representing citizens’ opinions, the dominant coalition was forced to talk to the minority coalition and solicit their opinions, which led to an agreement—the universal two-child policy.

Second, this study confirms Hypothesis 3. After the central authorities released relevant documents, various actors showed an obvious preferential attachment, which was a unique phenomenon of China’s policy change. From 2004 to 2009, compared with other actors, local governments showed an even more obvious preferential attachment to the central government. From 2010 to 2012, local governments enjoyed more autonomy in their discourse, showing a decrease in attachment to the central government. Simultaneously, experts played an increasingly important role in disseminating their policy position on the family planning policy through social media. With the popularisation of social media in China and the change in the CCP’s governing philosophy, the public became the most active group, pushing the change from 2013 to 2015, and an increasing number of citizens were participating in politics on the internet.

This study’s findings have notable implications in terms of theory, research design, and practice. Theoretically, this study attempted to modify the ACF, whose political context is different from that of the Western world. This attempt will help to further analyse the relative strengths and weaknesses of the ACF as a basis for understanding the policy process across political contexts. By introducing DNA, this study also provides a micro-perspective for revealing macro-policy change, which overcomes the obvious limitations of the ACF in explaining the dynamic change of coalition members. Network structures can quantify and visualise how policy belief changes promote the adjustment of coalition members. Empirically, the ERGM was used to show the preferential attachment phenomenon in China’s policy change process, which could be further studied. Practically, this article studies the micro-mechanism contributing to the change in China’s family planning policy, deepening the understanding of how policy changes in China’s context. Except for the central government, policy change also largely depends on multiple actors. However, in China, multiple actors cannot escape the impact imposed by the central authorities, and their preferences are deeply affected by the central authorities. On the surface, these actors can express their ideas and opinions towards a specific issue; however, their viewpoints are policy-oriented, which are largely restricted by policies issued by the authorities. Therefore, China’s policy change is shaped by interactions among multiple actors, but the central government still plays a leading role in this process.

In sum, by taking China’s family planning policy as an example, this study is dedicated to opening the ‘black box’ of China’s policy change. By introducing DNA, the flows of actors and policy beliefs were visualised. This study found that during the different policy stages, different actors formed different network structures of advocacy coalitions based on changing policy beliefs. The sharing and flow of policy beliefs among actors give rise to changes in network structures, which provides a mesoscopic perspective to understand China’s policy change. However, this study focuses on only the perspective of policy change in China’s family planning. Therefore, scholars interested in family planning may consider other perspectives or validity to broaden the range.

## Figures and Tables

**Figure 1 ijerph-20-05204-f001:**
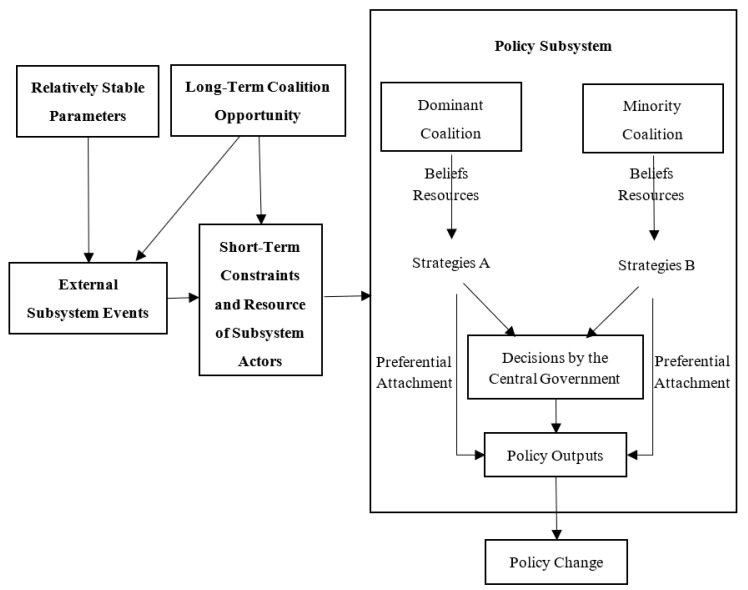
The Modified ACF.

**Figure 2 ijerph-20-05204-f002:**
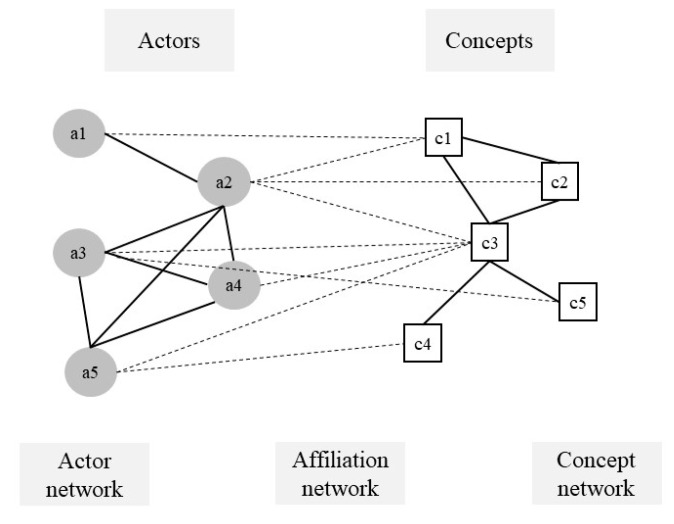
Discourse networks.

**Figure 3 ijerph-20-05204-f003:**
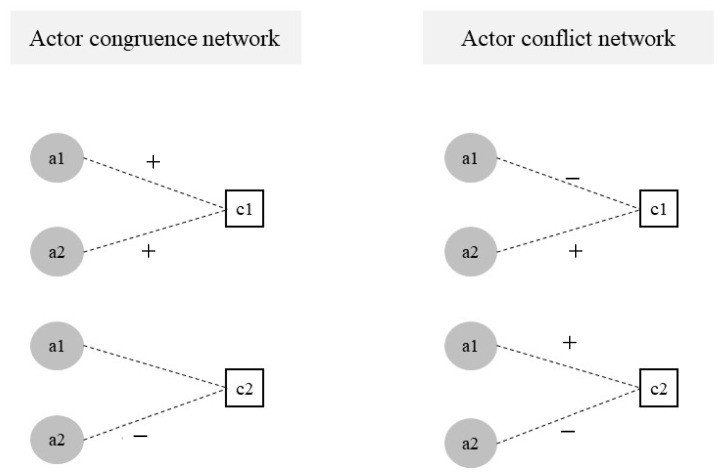
Two kinds of actor networks.

**Figure 4 ijerph-20-05204-f004:**
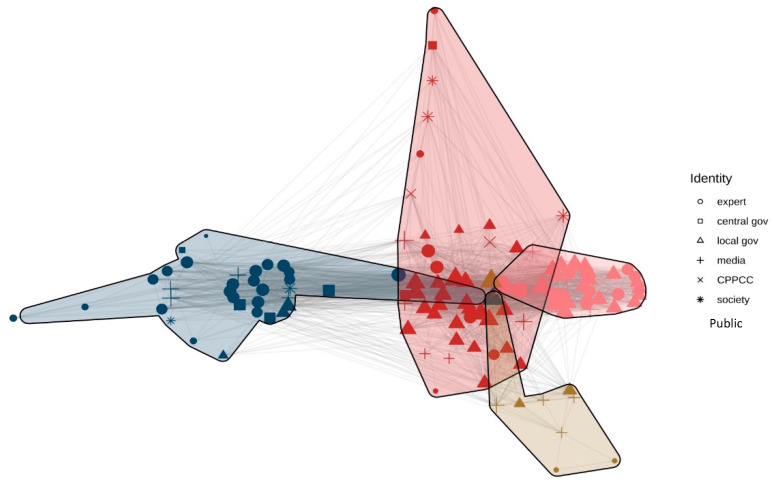
Discourse network visualisation (2004–2009). Note: Gov = government; CPPCC = Chinese People’s Political Consultative Conference; Red = dominant coalition; Blue = minority coalition; Yellow = the other coalition.

**Figure 5 ijerph-20-05204-f005:**
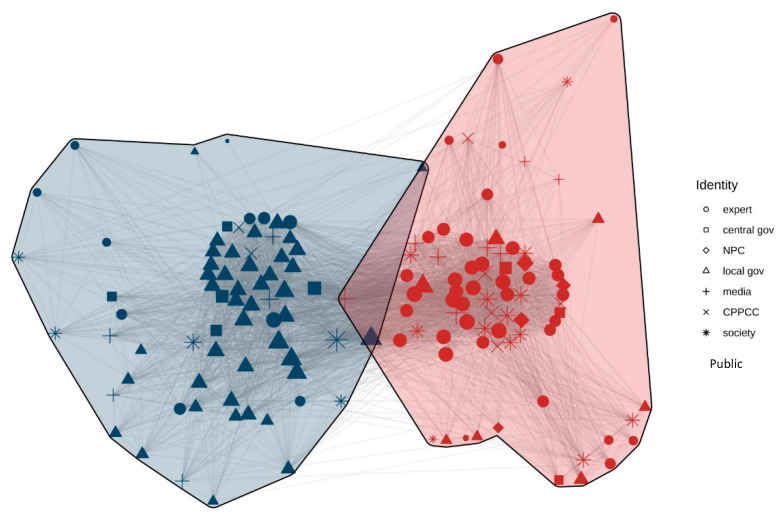
Discourse network visualisation (2010–2012). Note: NPC = National People’s Congress.

**Figure 6 ijerph-20-05204-f006:**
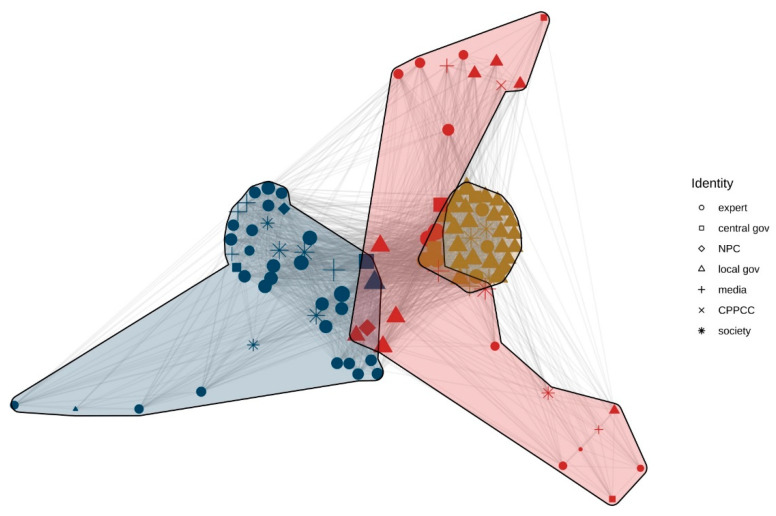
Discourse network visualisation (2013–2015).

**Table 1 ijerph-20-05204-t001:** Network factions density matrix (2004–2009).

	Dominant Coalition	Minority Coalition	Other Coalition
Dominant Coalition	0.873	0.174	0.072
Minority Coalition	0.174	0.722	0.104
Other Coalition	0.072	0.104	0.224

**Table 2 ijerph-20-05204-t002:** E-I index of networks (2004–2009).

	Number of Internal Relations	Number of External Relations	Total	E-I
Dominant Coalition	3052	495	3547	−0.721
Minority Coalition	616	400	1016	−0.213
Other Coalition	62	181	243	0.490
Overall Network				0.196

**Table 3 ijerph-20-05204-t003:** Degree centrality analysis of the overall network (2004–2009).

Actors	Degree	nDegree
Liaoning Government	187	0.166
Hu Chunhua	182	0.162
Jiangsu Government	176	0.156
Shandong Government	156	0.139
Zhang Weiqing	148	0.132
Wang Jineng	130	0.116
Han Changfu	119	0.106
Yang Qing	118	0.105
Pan GuiYu	117	0.104
Sun Shibin	113	0.100
National Population and Family Planning Commission	111	0.099
Du Qinglin	110	0.098
Wei Dongmei	103	0.092
Yang Xianming	101	0.090
Wang Jincai	99	0.088

**Table 4 ijerph-20-05204-t004:** Network factions density matrix (2010–2012).

	Dominant Coalition	Minority Coalition
Dominant Coalition	0.785	0.184
Minority Coalition	0.184	0.701

**Table 5 ijerph-20-05204-t005:** E-I index of networks (2010–2012).

	Number of Internal Relations	Number of External Relations	Total	E-I
Dominant Coalition	3388	831	4219	−0.606
Minority Coalition	2698	831	3529	−0.529
Overall Network	-	-	-	−0.571

**Table 6 ijerph-20-05204-t006:** E-I index of networks (2013–2015).

	Number of Internal Relations	Number of External Relations	Total	E-I
Dominant Coalition	118	336	454	0.480
MinorityCoalition	4646	336	4982	−0.865
Overall Network	−	−	−	0.417

**Table 7 ijerph-20-05204-t007:** Degree centrality analysis of discourse concept from the dominant coalition.

Dominant Coalition				
Policy Core Beliefs	Secondary Beliefs	Stage I	Stage II	Stage III
Seriousness of population problems	Large population base	0.086	0.051	0.041
	Large potential fertility population	0.030	0.000	0.004
	Fertility rate rebound	0.024	0.005	0.014
Causes of population problems	Financial burden	0.095	0.018	0.066
	Employment burden	0.009	0.005	0.000
	Education burden	0.000	0.003	0.008
	Environmental stress	0.102	0.049	0.042
	Traditional fertility concepts	0.032	0.013	0.000
	Illegal births by the rich	0.037	0.006	0.000
Advocating plans	The basic national policy	0.152	0.085	0.108
	Maintaining one-child policy	0.151	0.097	0.031
	Controlling the population size	0.039	0.012	0.013
	Improving the population quality	0.132	0.051	0.020
	One-child policy encouragement	0.175	0.078	0.026
	‘One vote’ veto system	0.082	0.044	0.015
	Improving the service level of family planning management	0.068	0.049	0.039
	Strengthening the leadership over the family planning work	0.142	0.043	0.016
	Improving family planning work methods	0.103	0.027	0.000
	Combination of government leading and villagers’ (residents’) autonomy	0.006	0.021	0.000
	Increasing family planning work funding	0.010	0.000	0.000
	Pre-potency propaganda	0.075	0.024	0.005
	Solving the family planning problem of floating population	0.060	0.034	0.004

**Table 8 ijerph-20-05204-t008:** Degree centrality analysis of discourse concept from the minority coalition.

Minority Coalition				
Policy Core Beliefs	Secondary Beliefs	Stage I	Stage II	Stage III
Seriousness of population problems	Low birth rate	0.047	0.069	0.063
	Disappearance of demographic dividend	0.065	0.059	0.070
	Ageing population	0.151	0.152	0.087
	Negative population growth	0.027	0.007	0.000
	Imbalance of sex ratios at birth	0.124	0.104	0.043
	Birth rate increasing insignificantly	0.023	0.031	0.058
	Adjusting the demographic structure	0.024	0.028	0.041
	The 4-2-1 family mode	0.033	0.021	0.062
Causes of population problems	Change in economic and social environment	0.060	0.055	0.029
	Change in marriage concept	0.046	0.034	0.044
	Calls for free birth rights	0.003	0.027	0.008
	Intricacy of bureaucratic system	0.000	0.020	0.000
	Losing only-child family	0.006	0.009	0.022
	Doting on the only child	0.017	0.011	0.003
Advocating plans	People-oriented	0.076	0.057	0.021
	Coordinating population issues	0.075	0.070	0.003
	Adjusting one-child policy	0.085	0.117	0.110
	Shuangdu Er’tai policy	0.075	0.051	0.005
	Dandu Er’tai policy	0.010	0.058	0.143
	Universal two-child policy	0.034	0.060	0.062
	No three-child	0.000	0.003	0.023
	Removal of restriction on the second childbirth interval	0.045	0.000	0.015
	Perfecting the policy of remarriage and fertility	0.047	0.000	0.004
	Maternity care for persons with special contributions	0.020	0.000	0.000
	Relaxing the fertility condition for families with disabled children	0.017	0.000	0.000
	Legalisation of ‘illegal’ two-child births	0.010	0.003	0.006
	Cancelling the one-child reward	0.000	0.002	0.005
	Abolishing one-child policy	0.000	0.007	0.005

**Table 9 ijerph-20-05204-t009:** Main effect model of discourse network.

		Estimate	Standard Error	Statistic
2004–2009	Edges	−1.149	0.070	0.033 *
	Local Governments	−0.198	0.040	<1 × 10^−4^ ***
	Expert	−0.486	0.043	<1 × 10^−4^ ***
	Media	−0.277	0.048	<1 × 10^−4^ ***
	Public	−0.660	0.074	<1 × 10^−4^ ***
AIC: 20,014; BIC: 20,053				
2010–2012	Edges	−0.147	0.059	0.012 *
	Local Governments	−0.224	0.034	<1 × 10^−4^ ***
	Expert	−0.333	0.035	<1 × 10^−4^ ***
	Media	−0.101	0.036	0.005 **
	Public	−0.660	0.074	0.031 *
AIC: 30,554; BIC: 30,595				
2013–2015	Edges	−0.146	0.079	0.065
	Local Governments	0.087	0.047	0.063
	Expert	−0.030	0.046	0.511
	Media	−0.063	0.059	0.287
	Public	0.177	0.055	0.001 **
AIC: 16,871; BIC: 16,908				

Note: * *p* < 0.05; ** *p* < 0.01; *** *p* < 0.001.

## Data Availability

Data set(s) associated with the manuscript will be available on request.

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
