# Peer review of "Policy Changes in China’s Family Planning: Perspectives of Advocacy Coalitions"

_ijerph, 2023, doi:10.3390/ijerph20065204_

Round 1
Reviewer 1 Report
This manuscript addresses an important topic – the role of advocacy coalitions in the reversal of China’s family planning policy, which finally ended the country’s 30-year One Child Policy. The authors particularly highlight the important role played by experts providing evidence-based arguments for shifting the policy. They also explain that despite this evidence, another group of local government officials were more persuasive to the Central Government and the National People’s Congress in maintaining the One Child Policy for an additional few years. With this said, I have a number of comments for the authors.
When I got to the discussion/conclusion, I found this very helpful – it explains how policy is made in China – an authoritarian regime. This is the context of China. Rather than framing the central government as part of a policy coalition, why not start with this – the central government clearly plays the dominant role in policymaking. What is the influence of advocacy coalitions in affecting policy change? The interplay between local governments and experts/media/public is interesting in and of itself.
“Practically, this article studies the micro-mechanism contributing to the change in China’s family planning policy, deepening the understanding of how policy changes in China’s context. Except for the central government, policy change also largely depends on multiple actors. However, in China, multiple actors cannot escape the impact imposed by the central authorities, and their preferences are deeply affected by the central authorities. On the surface, these actors can express their ideas and opinions towards a specific issue; however, their viewpoints are policy oriented, which are largely restricted by policies issued by the authorities. Therefore, China’s policy change is shaped by interactions among multiple actors, but the central government still plays a leading role in this process.
It would also help to have a clear methodology section followed by a results section. I found myself rereading the paper trying to figure out why a particular analysis (often introduced in the title of a table) was being done. A lot of terms are introduced but not really well explained. Was all of the analysis part of CFA and DNA? Actually, it looks like the analysis focused on DNA – how does CFA fit into the analysis? It seems the authors also added the ERGM – how is that related to CFA and/or DNA? To have all of the methodology together and then all of the findings together would help readers who are not statisticians or those emersed in CFA and DNA and all of the other analytic methods used in this paper. This manuscript is being reviewed for a journal of environmental research and public health, not public policy or political science. The last sentence is telling and gives a clear indication of the audience for this manuscript as currently written: “However, this study only focuses on one policy, China’s family planning. Therefore, scholars interested in applying the ACF to cases outside of Western democracies may consider other perspectives or validity to broaden the range.”
On the data – the manuscript says the data were from newspapers. That data source was supplemented by other evidence – like policy pronouncements, the expert study group, various articles and books. This use of supplementary material should be explained in the methods section. These other data sources are critical for giving context to the analysis of newspaper data.
This manuscript would benefit from inclusion of supplemental files of the analysis.
Specific comments:
Introduction: I was surprised to see only 3 references in the reference list that are about family planning in China – perhaps the title of the manuscript should be about modifying the CFA and using FP as an example. Either way, the manuscript would benefit from more references on family planning – including something about how the One Child Policy came about. Also, it would help to include references to the experts (all well-known demographers and others) and to the references that are in the text but not in the list of references.
“China's policy change in family planning has attracted increasing attention because of its rise as a superpower and its growing influence” [4]. Reference 4 is about one province in China. It isn’t about China as a superpower. And the world has been watching China’s One Child policy (not well received globally from a human rights perspective) which also helps explain the interest in the change in the policy.
“The advocacy coalition framework (hereinafter the ACF) was established to examine policy changes for over a decade” [8]. Sabatier says that policy change takes time so he included a time dimension - over a decade. The "for" in this sentence does not make sense.
Page 2: What is it about the ACF that makes it difficult to reflect micro-changes?
Page 2: “The ‘discursive turn’ of Western public policy research provides a method that overcomes the shortcomings of the ACF [17] by regarding discourse as the product of value intention and can change social reality.” What is the discursive turn? and what does it mean to regard discourse as the product of value intension - and what "can change social reality"?
Page 2: “It also provides a middle-range perspective to understand and analyse China’s large-scale policy change.” What does "middle-range perspective" mean? Is there a reference for it?
Page 2: “Although various studies have confirmed that ACF can be used to verify the policy change of authoritarian regimes [6,22,23], we still need an ACF, which is more suitable for China’s national conditions (see Figure 1).” What is it about China's national conditions that are not covered by ‘authoritarian regimes’? Also, the comma after ACF changes the meaning of the sentence. It should be removed.
Figure 1. It would be helpful to know what modifications were made in Figure 1 to make it more relevant for China? The components of the figure/ ACF needs to be explained.
Page 3: “However, unlike Western governments, which often act as intermediaries between coalitions, as a unitary political context, policy change exists in multiple governmental ministries and agencies relevant to China’s policy process [25]. Why is this not also the case in the West? Western governments are not monolithic - and in China - does policy change really happen in government ministries or in Party? National People's Congress and Standing Committee?
Page 3: In China, government departments, including the central and local government, hold the authority of public policy and occupy a core position [7].” Ref #7 is about policy coalitions. It is not sufficient here.
Page 3: “This belief system can be understood as a conceptual system that has three levels: deep core beliefs, policy core beliefs, and secondary beliefs.” This is not reflected in the figure and there is no explanation of where these three levels come from. Is there a reference for this?
Page 3: “However, once China’s major policies enter the implementation stage, due to the complexity of the national and social conditions, the interests of various localities, departments, organisations, groups, and even individuals will gradually solidify around the current policies, making it difficult to adjust deep core belief, let alone some untimely policies.” I don't understand this - if everyone knows they have to coalesce around policies - why does that make it difficult to adjust deep core beliefs? And what is the reference to "some untimely policies." What are those?
Page 4: “Once the objective situation is mature” – what does this mean?
Page 4: “Preferential attchment derives from graph theory, and it is related to the theory of Matthew effect [27,28].” How do these relate to the ACF and DNA introduced earlier? And what is the theory of Matthew effect?
Page 4: “Due to China’s top-down administrative hierarchy, multiple actors will move close to the central government inertially after a document is released.” I’m not sure inertially is the right word – and it is not a commonly used word. What is trying to be conveyed here?
Page 4: “local governments still have considerable autonomy” – explain what they have autonomy to do.
Page 4: “To use the ACF in China’s political system correctly, it is necessary to investigate the family planning policy, which exits at least two advocacy coalitions in the policy subsystem [32]:” – there are words missing is this sentence and the word exits isn’t correct.
Page 4 “The dynamic game between the two coalitions promotes the change and development of the family planning policy.” This sentence isn't clear - what tense? the recent policy change?
Page 4-5: “DNA establishes relationships among actors regarding policy beliefs and among actors who share the same policy beliefs belonging to an identical coalition.” This sentence is not clear - sharing policy beliefs and .... belonging to an identical coalition? What is an identical coalition?
Page 5: “China’s family planning policy was a process of centralisation and a typical example of an autocratic regime [36].” Does the reference (36) say that this is a typical example of an autocratic regime?
Page 5: “To some extent, policy change in China has become more open after the transition from a planned economy to a market economy, which is the combination of an authoritarian political system and transitional political context that makes it a valuable test to synthesise and understand the various applications of the modified ACF in China [22,25].” What is the transitional political context referred to in this sentence? And is it about the combination of authoritarian political systems and transitional political context that makes for more open policy change?
Page 5: “The extant research has indicated that newspapers are sources of multiple actors’ voices and that they reflect more than the views of the publishers [25,37].” How much state control is there over the media?
Page 5: what are “unmentioned concepts?” And “concepts that had not yet been developed”?
Figure 2: Is fig 2 based on real data? If so, label the actors and the concepts. Or say that it is an illustrative example.
Figure 3: the two sides look the same so it isn’t clear what the difference is between actor confluence networks and actor conflict network.
Page 7: ERGM – “The ERGM helps to examine the attribute characteristics of network nodes and the effect of the attribute characteristics on the network structure.” Please explain what ‘attribute characteristics’ are. Same with ‘network structure.’
Page 7: “The significance of using the ERGM is from being able to clarify the influence of multiple actors’ rights on the structure of the discourse network during different stages of policy change and present the micro-process of policy change. Specifically, this study used the main effect model of the ERGM through R.” What do ‘actors’ rights’ refer to here? Also see related comments linked to page 14 of the manuscript below on this.
Page 8: “Changes to the family planning policy were placed on the central government agenda.” Use active voice – who put it on the central government agenda?
Page 8: “What was the micro-basis of this policy change? This study divided the family planning policy into three stages: discourse diversion, bipolar discourse, and discourse confluence.” What does micro-basis mean?
Page 8: “In 2004, Professors Gu Baochang and Wang Feng established a research group on China’s fertility policy in the 21st century, and 18 population experts jointly drafted and signed a proposal to adjust the policy.” There must be a reference for this work. Also, both Gu Baochang and Wang Feng are well-known demographers – say who they are/their affiliations. This was not the first time the One Child policy was questioned. What was it about this attempt at coalition that made it on the policy radar? Do the authors know why Professors Gu Baochang and Wang Feng to ask them about why they formed this coalition in 2004 (and not earlier)?
Figure 4: What are the outer lines of the figure and what do the colors signify?
Figure 4: Need a key - what does CPPCC stand for. Also, use consistent terms - in the text it seems that "public" is used whereas in this figure it is "society"
Table 1: It is not intuitive how to interpret the numbers in the table. What does 0.873 mean?
Page 9: It is interesting that the authors list the names of famous experts but not government leaders - why is that?
Page 9: “degree of agglomerated subgroups” – it isn’t clear what this means.
How was density of subgroups measured? What data were used? Number of groups? Number of people? Number of internal vs. external relations?
Tables 2, 3, 5, and 6: The 3 decimal points are not needed. I suggest getting rid of them.
Page 9: The meaning of E-I is not intuitive - nor is the meaning of the last sentence – “The results in Table 2 are thus confirmed again.” I had to read this a few times to understand - how closely do groups clump together vs engage externally?
Page 9: “At this stage, experts from the fields of demography and sociology were the primary voices in the minority coalition. Many experts suggested their own views on adjusting the family planning policy, such as Professor Tian Xueyuan’s Three Population Forecasting Options, Professor Zhai Zhenwu’s Three-Step Plan, Professor Chen Youhua’s Four-Step Plan, and Professor Yi Fuxian’s famous Big Country with an Empty Nest.
What is the purpose of identifying the experts in the text if they are not deemed important in the Degree centrality analysis?
Also, why is the work of the experts mentioned in the text not included in the reference list?
What is the purpose of identifying the experts in the text if they are not deemed important in the Degree centrality analysis in Table 3?
Page 10: “Following research from 2005–2008, the research group on ‘China's fertility policy in the 21st century’ put forward Suggestions on the Adjustment of China's Population Policy in 2009. This proposal made some government actors change their attitudes and reflect on the applicability of the one-child policy, and it stimulated an increasing number of actors to participate in the policy debate.” How do the authors know this? From your analysis or another source? Please explain and add references, as appropriate.
Figure 5. If the media fall off "other coalition" - why are they displayed in Fig 5?
Page 10: “There are only two advocacy coalitions on behalf of the minority and the dominant coalitions, respectively, while the ‘other coalition’ subgroup disappears. A possible explanation is that some actors held a wait-and-see attitude from 2004–2009, formed their policy beliefs through policy learning over time, and joined the debate of the two major advocacy coalitions from 2010–2012.” But if the media are in "other" how can they then switch to being experts - joined with the expert coalition?
Page 11: “However, at this stage, the dominant coalition, composed of local governments, op[1]posed the loosening of the family planning policy. For example, the local population and family planning committees in many provinces and cities held negative opinions on slack[1]ening the fertility policy because they were worried about the rebound of the fertility rate and the pressures on limited resources and environmental pollution. With the opposition voice being stronger, the process of policy change was set aside on account of the convening of the 18th National Congress of the CCP in 2012 and the subsequent government institution reforms of the State Council. The minority coalition and dominant coalition thus entered a state of confrontation.” This is good, but it gets buried under so much analysis. Was all this analysis really needed to make this point that seems obvious. And why would it be surprising that the central government/Party would listen to the provincial and city governments over listening to experts, the media and the public?
Page 11: In March 2013, the National Population and Family Planning Commission and Min[1]istry of Health merged to establish the National Health and Family Planning Commission of the People's Republic of China in the new round of government institutional reforms of the State Council. The change in organisational structure released the policy change signal.” The wording – ‘released the policy change signal’ is awkward. I suggest – ‘signaled the shift in policy.’ Also, was there any acknowledgment on the part of the dominant coalition of the views of the minority coalition?
Figure 6: Why is there a third circle in this figure when there are only 2 advocacy coalitions?
Page 12: “Table 6 shows the E-I index analysis. The E-I index of the dominant coalition is 0.480, and the external relationship index is larger than the internal relationship index Moreover,…” Need a period between index and Moreover
Page 12: “During the 2013 National People’s Congress and Chinese People’s Political Consultative Conference, some representatives proposed gradually loosening the family planning policy.” REVIEWER: Who was it that suggested the loosening of the policy?
Page 12: “In 2014, a joint proposal letter from 5,000 couples asking for the universal two-child policy became a landmark event in the national attitude on this topic.” Who orchestrated this letter from 5,000 couples and for what purpose? Is there a reference for it?
Tables 7 and 8 are hard to take in - what the numbers under states I, II and III mean is not intuitive, so it is hard to follow the discussion that follows these tables and to see where the authors are getting the points that they are making.
Tables 7 and 8: Could there be some shading or something to show the important findings in tables 7 and 8
Page 14: “(3) At the discourse confluence stage, the degree centrality of the discourse concept of the minority coalition was generally higher than that of the dominant coalition.” ‘Degree centrality of the discourse concept’ - what does that mean? How are the 28 policy beliefs related to that?
Page 14: “Preferential attchment in policy change” – what does ‘preferential attchment’ mean?
Page 14: “Due to the presupposition of the ACF, the right dimension of multiple discourse actors had not been realised.” This sentence is incomprehensible. What is the "presupposition of the ACF" What is the "right dimension of multiple discourse actors" Does right mean "correct," "right side," “rights of” - either way, it isn't clear what it means. See related comment from page 7 above.
Page 15: “The public’s response and support for central policy became the strongest voice in the discourse network. However, the direction of the family planning policy had been very clear because of the implementation of the ‘policy boots’; therefore, apart from the public, there was no need for more actors to speak for the adjustment of family planning policy, and the connection to the central government was no longer significant.” How controlled was the public's response to the policy? Did the government allow what it wanted to hear to be published?
Page 15: “The policy beliefs held by actors outside the government proposed the policy change and put it on the government’s agenda.” Outsider actors can't put something on the government's agenda - only the government can do that. Do you mean that their advocacy resulted in it being put on the government's agenda?
Page 16: “For the third stage (2013–2015), due to increasing pressure posed by the minority coalition, the dominant coalition was enforced to talk to the minority coalition which led to an agreement – the universal two-child policy.” What is the evidence of this ‘pressure’? Weren't they just doing evidence-informed advocacy? What was the ‘pressure’ part of it? Was it just that their evidence was becoming increasingly clear about the detrimental effect of continued implementation of the one-child policy on the country? Use of the term ‘enforced’ is incorrect – do you mean they were forced to talk to the minority coalition? In what way were they forced?
Page 16: “Simultaneously, experts played an increasingly important role in changing the family planning policy through social media.” They didn't change the policy through social media. They used social media to disseminate their policy position.
Page 16: “With the popularisation of social media in China and the change in the CCP’s governing philosophy, the public became the most active group, pushing the change from 2013–2015, and an increasing number of netizens were participating in politics in a new way.” Why is social media and "netizens" only coming up in the discussion and conclusion? If they are an important part of an advocacy coalition, they should be discussed in the results section.
Page 16: “By introducing DNA, this study also provides a micro-perspective for revealing macro-policy change, which overcomes the obvious limitations of the ACF in explaining the dynamic change of coalition members.” What are the ‘obvious limitations’?
Page 16: “The sharing and flow of policy beliefs among actors gives rise to changes in network structures, which provides a middle-range perspective to understand China’s policy change.” Since middle-range was not explained earlier (see comment), it isn’t clear here what it means.
Reviewer 2 Report
I have made a VERY FEW suggestions on the PDF version of the paper, and I attach my suggestions here.
My only substantive question/concern/comment deals with the unauthorized practice under one-child policy when parents-to-be terminated a pregnancy if the fetus was female so that their one child would be male. That, I believe, was the practice that led to imbalance in sex ratio.
Is that subject taboo?

Author Response
Dear Reviewer 2,
Thank you very much for your time and your work on our paper, “Policy Changes in China’s Family Planning: A Perspectives of Advocacy Coalitions” (Manuscript ID: ijerph-2113763). My co-authors and I are very grateful for your message and the comments.
We have substantially revised our manuscript based on those comments and suggestions. Attached please find a summary of the revisions we have made in the new manuscript in response to the reviewers’ insightful comments. For better display, we submit the revised version with track changes.
We think the new version has been much improved and that our changes have addressed most of the concerns raised by all reviewers. We look forward to learning of your reactions to our revisions. Thank you once again!
Yours sincerely,
Bojia Liu
Considering the revisions covers the full text and is relatively scattered, we will give an overall description of the revised content of the article, and then reply to reviewers’ comments point by point.
Overall, we have made the following modifications:
- Improved the reference in the manuscript.
According to the suggestions of the reviewers, we have made a more detailed explanation in different parts of the manuscript with new evidence by supplementing related literature reading and strengthening theoretical discussion.
- Improved the expressions and the results analysis.
According to the comments of the reviewer, we have modified some of the expressions in the manuscript and the figures with some wrong or missing symbols. In addition, to make the results more directly connected to the figures and tables, we have added some explanations in the paragraphs.
- Adjustment and improvement of other details.
It mainly includes: First, we repeatedly proofread the revised version and try to avoid the problems of inappropriate expression, typography, reference format, punctuation, page number, formula model, etc.; Secondly, we have revised and polished the language based on the reviewers’ suggestion in the manuscript.
Next, we will respond to the comments put forward by the reviewer point by point:
Reviewer#2, Comment # 1: I have made a VERY FEW suggestions on the PDF version of the paper
Author response: Many thanks to Reviewer#2 for the time spent reviewing this manuscript. After reading Reviewer#2’s detailed suggestions of the manuscript, we have adjusted the expressions based on these suggestions accordingly.
Reviewer#2, Comment # 2: My only substantive question/concern/comment deals with the unauthorized practice under one-child policy when parents-to-be terminated a pregnancy if the fetus was female so that their one child would be male. That, I believe, was the practice that led to imbalance in sex ratio.
Author response: This is a very helpful comment and we thank Reviewer#2 very much for pointing this out. We would like to make the following explanation.
Author action: While the sex ratio distortion in China is a long-standing demographic pattern, fertility policies instituted to slow population growth have exacerbated the female deficit. Under the One-Child Policy, parents in China who exceed their fertility limit are forced to pay a fine and are subject to a variety of other monetary punishments, including the seizure of property and forced dismissal from government employment. Gender preference used to increase the fertility rate in a period, as people in China sought at least one male descendant by having extra births. However, the development of sex-identification and abortion technology has led to less influence of the preference for a son on birth numbers and the fertility rate, as people realize their son preference by sex-selective abortion rather than having extra births. Therefore, China’s fertility policy has tightened partially due to the sex ratio distortion.
Although rapid industrialization and large changes in fertility have reshaped China in the last 40 years, sex preferences have survived the transition. In an earlier era of high fertility, they were manifested in higher stopping probabilities following sons and had a muted effect on the overall sex ratio. Today, fertility in China has slowed but the imbalance in the sex ratio has become a pressing concern. Therefore, encouraging or forcing people to change their fertility behavior without addressing their fundamental preferences may have unanticipated consequences to the society due to the development of sex-identification and abortion. And in recent years, the relaxation in the One-Child Policy could reduce the sex ratio at birth by allowing more parents to have a son without resorting to sex selection.
Finally, thanks again for Reviewer#2’s careful reading and valuable suggestions. Due to our limited ability, there may have still some defects in the revised manuscript, and not all of the specific comments reflect in the response letter specifically. We hope the revised manuscript combined with the response letter will give a better display and address most of the concerns compared to the original version.
Round 2
Reviewer 1 Report
This manuscript is improved and clearer. I have a few additional comments.
Pages 3-4. “However, once China’s major policies enter the implementation stage, due to the complexity of the national and social conditions, the interests of various localities, departments, organisations, groups, and even individuals will gradually solidify around the current policies, making it difficult for the specific interest groups to adjust deep core belief, let alone to adjust some untimely policies like family planning policy.” The phrase “adjust some untimely policies like family planning policy” is not clear. What is “untimely” about the family planning policy in this context?
Page 4. “Once the objective situation of birth control is mature or the population problems become serious enough to threaten the regime survival, the central government will promptly adjust the policy, break the original stable policy, and change its deep core beliefs [26].” The phrase “objective situation of birth control is mature” is awkward. So is the phrase “or the population problems become serious enough to threaten the regime survival” – could both of these be phrased more clearly.
Page 4. “Preferential attachment derives from graph theory, and it is related to the theory of Matthew effect that the gap between good and poor actors increases with time [27,28]. What is graph theory? What is the theory of the Matthew effect? What is meant by “good” and “poor” actors?
Page 4. “The dynamic game between the two coalitions promoted the change and development of the family planning policy.” Do you mean promoted the need for change in the family planning policy and development of a new policy?
Page 7. ERGM. “The network structure effects refer to ’endogenous structural factors’ which are only from the internal processes of networks, such as the number of network edges, mutuality, convergence. The network structure enable ERGM to reveal the role of the observed specific dependency structure in the network formation process. For attribute characteristics, we need to consider variables which includes basic demographics, economic factors, social environments, and so on.” This explanation is not comprehensible to a lay reader.
Page 8. “The minority coalition emerged after a period of research undertaken and policy oriented undergone by National Family Planning Commission expert advisors.” There are words missing in this sentence. Do you mean a period of policy-oriented research undertaken by external experts and provided to the National Family Planning Commission? Or is this research that the NFPC requested from experts? Were the experts affiliated with the NFPC?
Page 9. “According to Figure 4 and Table 1, there are were three advocacy coalitions from 2004–2009. Tian Xueyuan, Zeng Yi, Zhai Zhenwu, and other experts represent the minority coalitions.” Why is minority coalition plural? Above this, Professors Gu Baochang and Wang Feng are mentioned, but they are not include here as part of the minority coalition. Why not? And please tell the reader who Tian Xueyuan, Zeng Yi, and Zhai Zhenwu are. They are only said to be “experts” – what kind of experts? I see you had this detail on page 10 but deleted it. I think it is helpful for readers.
Page 12. Why was the example about the local population and family planning committees deleted? That is helpful for readers.
Page 16. The term “policy boots” is not clear.
Page 18. I don’t understand the editing in the final sentence. Don’t the authors want to say that they have applied the modified ACF to family planning policy in China and that other scholars could use it to assess other policies – in China and elsewhere. Or are the authors saying that other scholars can use other analytic methods to study the change in family planning policy in China.
Centrality analysis (tables 3, 7 and 8) – the numbers in the tables are still not intuitive. The explanations added before these tables aren’t enough for the reader to interpret the numbers in the tables.
